cognition/neuroscience

human evolution, symbols, engravings, perception, ventral pathway, functional magnetic resonance imaging

**Author for correspondence:**
E. Mellet
e-mail: emmanuel.mellet@u-bordeaux.fr

# Neuroimaging supports the representational nature of the earliest human engravings

E. Mellet[1,2,3], M. Salagnon[1,2,3], A. Majkić[4], S. Cremona[1,2,3], M. Joliot[1,2,3], G. Jobard[1,2,3], B. Mazoyer[1,2,3], N. Tzourio Mazoyer[1,2,3] and F. d'Errico[4,5]

[1]Institut des Maladies Neurodégénératives, UMR 5293, Groupe d'Imagerie Neurofonctionnelle, Université de Bordeaux, 33000 Bordeaux France
[2]CNRS, GIN, IMN UMR 5293, Bordeaux, France
[3]CEA, GIN, IMN UMR 5293, Bordeaux, France
[4]PACEA UMR 5199, University Bordeaux, CNRS, Pessac, France
[5]SFF Centre for Early Sapiens Behaviour (SapienCE), University of Bergen, Bergen, Norway

EM, 0000-0002-2676-9112; SC, 0000-0002-2294-8199; FdE, 0000-0002-2422-3079

The earliest human graphic productions, consisting of abstract patterns engraved on a variety of media, date to the Lower and Middle Palaeolithic. They are associated with anatomically modern and archaic hominins. The nature and significance of these engravings are still under question. To address this issue, we used functional magnetic resonance imaging to compare brain activations triggered by the perception of engraved patterns dating between 540 000 and 30 000 years before the present with those elicited by the perception of scenes, objects, symbol-like characters and written words. The perception of the engravings bilaterally activated regions along the ventral route in a pattern similar to that activated by the perception of objects, suggesting that these graphic productions are processed as organized visual representations in the brain. Moreover, the perception of the engravings led to a leftward activation of the visual word form area. These results support the hypothesis that these engravings have the visual properties of meaningful representations in present-day humans, and could have served such purpose in early modern humans and archaic hominins.

# 1. Rationale and results

From Palaeolithic rock paintings to contemporary art, the production and perception of symbolic artefacts have represented a major aspect of human cognitive activity.

However, no consensus exists on when, how and among which of our fossil ancestors symbolically mediated behaviour first arose. The ability to embed meaning in cultural products was for long considered the result of a sudden cognitive revolution occurring among Modern Human populations settling in Europe 42 000 years ago and replacing the resident Neanderthals. The cultural complexity of these populations, demonstrated by their mastery in painting, drawing, carving and the sophistication of their clothing, body ornamentation and mortuary practices was taken as the self-evident proof of that cognitive revolution [1–3].

The subsequent discovery at older African sites of artefacts—modified ochre, beads, drawings, engravings, primary burials—has led many authors to propose that symbolic practices emerged on that continent well before Modern Human arrival in Eurasia, and as a direct consequence of the African origin of our species 200 000 years ago, involving the emergence of our modern cognition [4].

The direct connection between the possible earliest instances of symbolic behaviour in Africa and the origin of our species has been challenged on multiple grounds. Some authors have argued that artefacts suggesting symbolic practices appear in Africa over a very long period of time, and are not found in many regions until a few thousand years ago, which supports scenarios in which symbolic behaviour spreads from local cultural trajectories rather than a single speciation event [5]. Others have noticed that a modern humans–modern cognition equation is contradicted by the fact that prior to modern human dispersal out of Africa, comparable symbolic practices existed in Eurasia among archaic populations such as the Neanderthals [6–10]. Not all researchers, however, are willing to grant a symbolic dimension to old African and Eurasian artefacts interpreted by some as the archetypes of our modern, fully symbolic, cognition and material culture. Mineral pigment, whose use goes back in Africa and Europe to at least 300 ka [11,12], has been considered as an ambiguous proxy of symbolic thinking on the ground that iron oxides may have also been used for utilitarian functions such as sunscreen or an additive for the preparation of mastic [13,14]. Personal ornaments, attested in Africa since at least 120 ka [15,16], are generally considered a technology specific to humans, which signals their ability to project information to members of the same or neighbouring groups by a shared symbolic language [17,18]. Some have, however, contended that early beads may only reflect attention to personal identity and do not necessarily stand for something else [19]. Others have argued that beads cannot be taken as reliable archaeological indicators of a complex language because the inferential steps on which the bead-language equation is based does not explain why a complex language is necessary for transmitting the symbolic meaning associated with the use of ornaments [20]. Abstract paintings, engravings and drawings older than 42 ka, attributed to *Homo erectus* [21], *Homo neanderthalensis* [22,23] and early African *Homo sapiens* [24], are taken by some as compelling evidence that they were used symbolically, i.e. to communicate a meaning distinct from their possible iconic referent [25]. This view is reinforced by detailed analyses showing that these representations were produced deliberately and had no apparent utilitarian function [21,22,26,27]. Researchers reluctant to grant symbolic status to these representations argued that they are too rare [28], not well enough dated [29], or that they only represent an intermediate step toward symbolism. Hodgson, for example, has proposed that since the primary visual cortex privileges the recognition of line junctions over other natural visual stimuli, the earliest graphic representations simply mirrored the elemental topological features that the early visual cortex was primed to process [30,31]. The motivation behind their production would have only been aesthetic or 'proto-aesthetic' and their perception would have only elicited the primary visual cortex. However, this and other theories concerned with the emergence of symbolic behaviour and its impact on human cognition have not been the subject of empirical research.

The present study aims to characterize the cerebral regions involved in the perception of these early engravings. This work complements recent attempts to apply neuroimaging techniques to test hypotheses on the evolution of cognitive functions. Although mapping modern human brains has intrinsic limitations to draw definitive conclusions on past brain organization, this approach has proved useful in studying the relationship between cognitive networks involved in the coevolution of tool-making and language functions [32–34]. We report here the first attempt to shed light on the function of Palaeolithic engravings by mapping the brain regions involved in their perception.

The blood oxygen level-dependent (BOLD) signal was mapped in 27 healthy volunteers using functional magnetic resonance imaging (fMRI), while the individuals were presented with tracings of

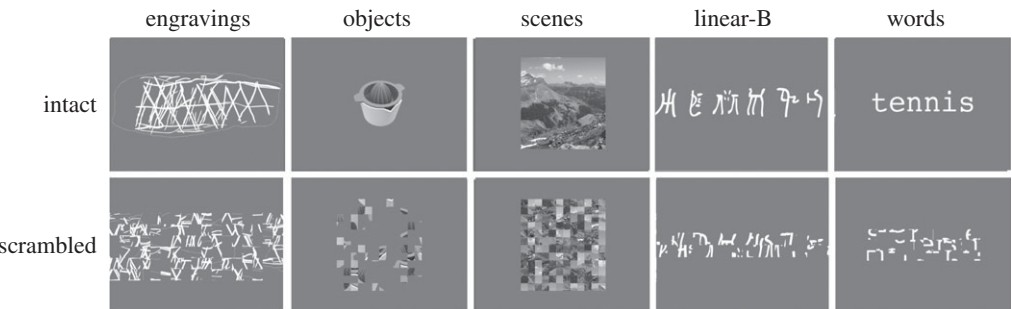

**Figure 1.** Examples of intact and scrambled stimuli used in the 1-back task.

engravings dating between 540 ka and 30 ka (see electronic supplementary material, table S1). The perception of these patterns was compared to the perception of their scrambled version, in which the organization of the abstract patterns is lost, to assess whether the geometric organization of the engravings (albeit some patterns were very simple) could be differentiated at the visual cortex level from patterns with no perceptual organization. The relatively simple organization that characterizes the earliest engravings may be perceived as not being sufficiently different from the scrambled version and may therefore not engage differently brain visual regions. In this case, no activation will be highlighted by the 'engravings minus scrambled engravings' contrast. Another possibility is that the difference does not involve the ventral cortex but just the primary visual area. This cortical region has been hypothesized to have played a crucial role in the production of the earliest engravings [30,31]. The second aim of the study was to shed light on the nature of the engravings using the observed activation pattern. To achieve this goal, we assessed whether and to what extent the regions activated by the perception of the ancient engravings were also involved in the processing of other visual categories and their scrambled version (figure 1).

The same participants were presented with distinct visual categories. The first category included pictures of nameable human-made artefacts. These stimuli conveyed semantic and lexical information but no symbolic content. The rationale behind this choice is that proximity between patterns of activation of the engravings and objects would support the representational nature of the engravings, which could also indicate that the engravings were processed as a whole, rather than a random combination of lines, and fulfilled the 'good shape' criteria of the gestalt theory of perception [35,36]. The second category was represented by outdoor scenes depicting large-scale surroundings. These stimuli were chosen to convey information regarding the outer world and semantic content without being perceived as a single item. Two supplementary symbolic categories were included in the study. Chains of characters from the linear-B syllabic script, which is a writing system unknown to the participants, were chosen as stimuli perceptible as 'potentially' symbolic. A similar profile of activation for engravings and linear-B could suggest that the former are processed as symbols or signs consisting of combinations of simple elements. The last symbolic category was represented by two-syllable words. This condition extended the requirements of linear-B characters because the perception of familiar overlearned signs (letters) and their familiar combinations trigger access to the pronunciation and meaning of words. In addition, word reading is characterized by a leftward asymmetric activation of the ventral route that reflects symbol processing and lexical and semantic access [37,38]. Thus, this condition provides a pattern of leftward asymmetry that could be contrasted with the asymmetry, if any, elicited by the perception of the engravings. The regional BOLD values in response to the engravings *minus* the scramble contrast were extracted from each homotopic region (hROI) of the functional homotopic atlas AICHA [39]. The regions included in the analysis fulfilled the following two criteria: in at least one hemisphere a significant positive value was observed in response to both the engravings *minus* scramble contrast ($p < 0.05$ false discovery rate (FDR) corrected, one-sample $t$-test) and engravings *minus* fixation (see methods) contrast ($p < 0.05$ uncorrected, one-sample $t$-test). The fulfilment of these conditions ensured that regions with stronger deactivation under the reference condition (scramble) during the engravings comparison with the fixation were not selected.

Ten hROIs were significantly activated in either the right or left hemisphere in the engravings *minus* scramble contrast. All hROIs were located in the visual ventral pathway (table 1 and figure 2; electronic supplementary material, table S2). The perceptual processing of the engravings involved visual regions beyond the primary visual and peristriate areas, while primary visual areas were activated similarly by both intact and scrambled versions of engravings. Previous work has led authors to propose that the

**Table 1.** Mean BOLD value of the 10 hROIs activated in the engraving *minus* scramble contrast and left *minus* right asymmetry under the five conditions. Colour code: pink, significant activation; blue, significant deactivation; orange, significant left asymmetry; green, significant right asymmetry. All tests: one-sample test, $p < 0.05$ corrected.

| | OLat3 | O2-2 | O3-2 | O3-1 | T3-5 | FUS5 | T3-4 | FUS4 | FUS3 | FUS2 |
|---|---|---|---|---|---|---|---|---|---|---|
| **engravings** | | | | | | | | | | |
| BOLD LH | 0.25 (0.11) | 0.16 (0.06) | 0.46 (0.09) | 0.49 (0.09) | 0.27 (0.07) | 0.20 (0.05) | 0.31 (0.08) | 0.39 (0.08) | 0.18 (0.04) | 0.13 (0.05) |
| BOLD RH | 0.45 (0.16) | 0.17 (0.07) | 0.43 (0.10) | 0.36 (0.10) | 0.16 (0.06) | 0.25 (0.07) | 0.19 (0.11) | 0.28 (0.09) | 0.13 (0.04) | 0.12 (0.04) |
| asym L-R | −0.20 (0.09) | 0.00 (0.03) | 0.03 (0.06) | 0.13 (0.06) | 0.11 (0.04) | −0.05 (0.04) | 0.13 (0.06) | 0.12 (0.04) | 0.05 (0.02) | 0.00 (0.03) |
| **objects** | | | | | | | | | | |
| BOLD LH | 0.37 (0.08) | 0.11 (0.05) | 0.52 (0.07) | 0.55 (0.08) | 0.26 (0.05) | 0.34 (0.04) | 0.37 (0.07) | 0.43 (0.07) | 0.19 (0.04) | 0.19 (0.03) |
| BOLD RH | 0.48 (0.10) | 0.15 (0.05) | 0.41 (0.06) | 0.16 (0.07) | 0.12 (0.05) | 0.25 (0.04) | −0.06 (0.07) | 0.25 (0.05) | 0.10 (0.03) | 0.17 (0.03) |
| asym L-R | −0.11 (0.08) | −0.04 (0.03) | 0.11 (0.05) | 0.40 (0.07) | 0.13 (0.04) | 0.08 (0.03) | 0.43 (0.04) | 0.19 (0.05) | 0.09 (0.02) | 0.02 (0.03) |
| **words** | | | | | | | | | | |
| BOLD LH | −0.22 (0.06) | −0.41 (0.05) | −0.28 (0.06) | −0.47 (0.08) | −0.22 (0.05) | −0.18 (0.03) | −0.08 (0.07) | 0.25 (0.06) | 0.07 (0.04) | 0.07 (0.03) |
| BOLD RH | −0.43 (0.08) | −0.36 (0.06) | −0.45 (0.07) | −0.66 (0.06) | −0.23 (0.04) | −0.27 (0.04) | −0.58 (0.08) | −0.16 (0.05) | −0.07 (0.03) | 0.04 (0.03) |
| asym L-R | 0.21 (0.06) | −0.04 (0.04) | 0.17 (0.04) | 0.19 (0.08) | 0.01 (0.06) | 0.10 (0.04) | 0.50 (0.08) | 0.41 (0.04) | 0.14 (0.03) | 0.11 (0.03) |
| **linear-B** | | | | | | | | | | |
| BOLD LH | 0.32 (0.08) | −0.02 (0.05) | 0.33 (0.07) | 0.19 (0.09) | 0.06 (0.06) | 0.14 (0.04) | 0.14 (0.07) | 0.40 (0.07) | 0.14 (0.04) | 0.14 (0.04) |
| BOLD RH | 0.42 (0.10) | 0.01 (0.04) | 0.26 (0.06) | −0.01 (0.08) | 0.01 (0.06) | 0.04 (0.04) | −0.12 (0.07) | 0.23 (0.04) | 0.11 (0.03) | 0.10 (0.03) |
| asym L-R | −0.10 (0.09) | −0.03 (0.04) | 0.06 (0.04) | 0.20 (0.06) | 0.05 (0.05) | 0.10 (0.04) | 0.25 (0.05) | 0.17 (0.06) | 0.03 (0.02) | 0.04 (0.03) |
| **scenes** | | | | | | | | | | |
| BOLD LH | −0.02 (0.06) | 0.21 (0.04) | 0.12 (0.04) | 0.07 (0.05) | 0.06 (0.03) | 0.31 (0.04) | 0.07 (0.05) | 0.10 (0.05) | 0.06 (0.03) | 0.07 (0.02) |
| BOLD RH | 0.14 (0.06) | 0.20 (0.04) | 0.15 (0.05) | 0.00 (0.05) | 0.09 (0.03) | 0.44 (0.05) | −0.07 (0.06) | 0.11 (0.04) | 0.10 (0.03) | 0.12 (0.02) |
| asym L-R | −0.16 (0.03) | 0.01 (0.03) | −0.04 (0.02) | 0.07 (0.03) | −0.04 (0.02) | −0.13 (0.04) | 0.14 (0.03) | −0.01 (0.03) | −0.04 (0.02) | −0.05 (0.02) |

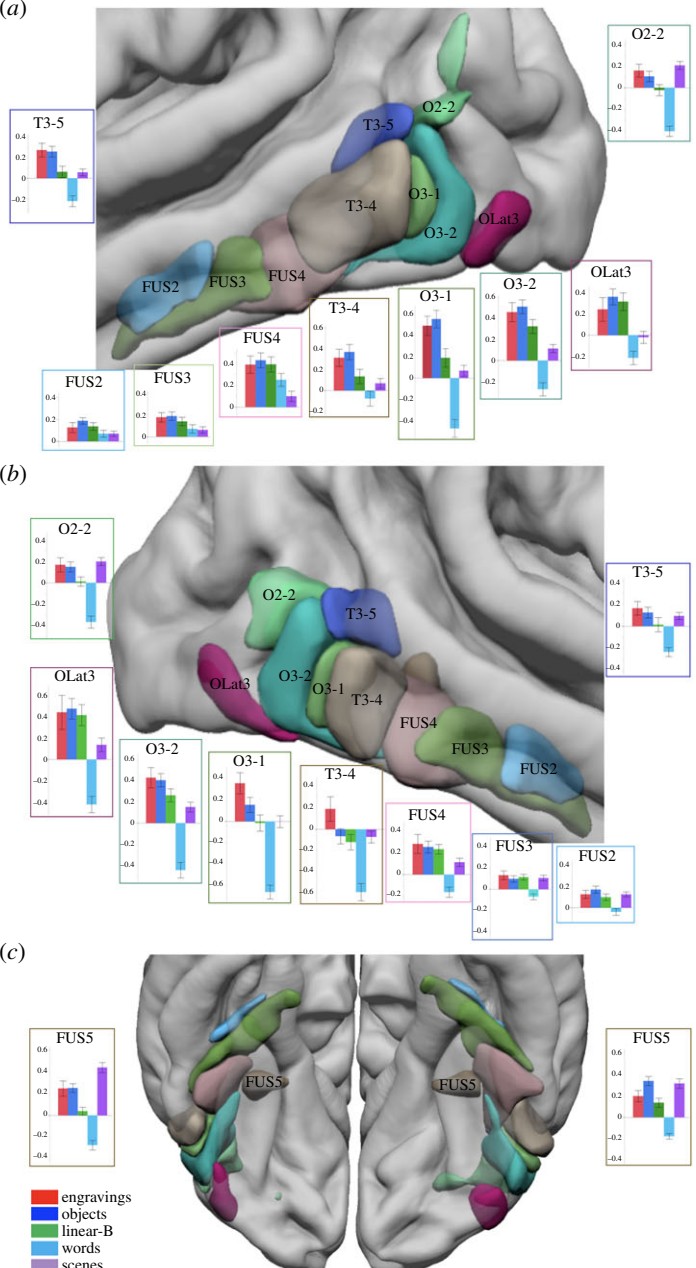

**Figure 2.** Superimposition on an MRI template of the 10 hROIs showing a significant BOLD signal increase in the engravings *minus* scramble contrast. (*a*) Lateral view of the left hemisphere, (*b*) lateral view of the right hemisphere and (*c*) inferior view of the left and right hemispheres. Bar plots represent the BOLD values obtained in the five categories *minus* their scrambled version contrasts in each region. Error bars represent the standard error of the mean.

early visual cortex played a key role in the extraction of the geometric features composing the ancient engravings [30,40]. Our results show instead that the activity generated by the perception of scrambled and intact engravings in the early visual cortex does not substantially differ. Thus, they do not support the statement according to which the primary visual cortex was specifically involved in the processing of the structural properties of the engraved patterns. Rather, they emphasize the role of more anterior visual regions belonging to the occipito-temporal pathway in the perceptual processing of the engravings.

Although the engravings are not recognized as existing entities, their perception elicits activation in anterior regions of the ventral pathway sensitive to distinct visual categories [41–44]. The involvement of the visual ventral pathway indicates that information processing requires more than the mere identification of visual primitives, such as the simple extraction of edges, oriented segments or ends of

lines. This implies that the level of perceptual organization characterizing the engravings is sufficiently high to recruit higher-order visual areas.

Interestingly, in the left hemisphere, the 10 regions activated in the engravings *minus* scramble contrast were also significantly activated by the perception of objects compared to their scrambled version (table 1 and figure 2). Repeated-measures ANOVA showed that there was no interaction effect between the concerned hROIs, engravings and object perception conditions on the BOLD values ($F_{9,225} = 1.6$, $p = 0.11$), which reflect the similarity of activation of these two conditions in the left hemisphere. In the right hemisphere, a condition by region interaction was present ($F_{9,225} = 2.4$, $p = 0.02$) due to a significantly lower BOLD value in the right inferior temporal hROI (T3-4) under the object condition than under the engravings condition. No other regions showed a significant BOLD difference between these two conditions in the right hemisphere. The activation profile in response to each visual category compared to its scrambled version among the 10 hROIs confirms that the engravings and object conditions triggered similar activations along the ventral pathway (figure 3). Notably, the activation profile under the engravings perception condition in the 16 hROIs activated by object perception (listed in electronic supplementary material, table S3) was also identical (see electronic supplementary material, figure S1), providing supplementary evidence of the brain functional overlap between objects and engravings perception.

By contrast, the comparison of the BOLD values observed under the engravings condition with those in each of the other conditions revealed strongly significant hROIs × conditions interactions in all cases in both the left (engravings, linear-B ($F_{9,225} = 3.6$, $p = 0.0003$), engravings, words ($F_{9,225} = 23.6$, $p < 0.0001$), engravings, scenes ($F_{9,225} = 13.9$, $p < 0.0001$)) and the right (engravings, linear-B ($F_{9,225} = 3.4$, $p = 0.0006$), engravings, words ($F_{9,225} = 18.0$, $p < 0.0001$), engravings, scenes ($F_{9,225} = 11.4$, $p < 0.0001$)) hemispheres. Thus, the activation profile under the linear-B script, words and scenes conditions compared to their scrambled version differed from that under the engravings condition, although some regions exhibited the same level of activation (figure 3*a*). The plot of activations under the engravings condition against the activations under each of the other conditions suggests that a positive linear relationship exists only between the objects and engravings conditions (figure 3*b*).

To verify that the relationship observed at the group level between the activation profiles elicited by objects and engravings was not determined by outliers, the correlation of their activation profiles in 10 regions, compared to their scrambled version, was computed in each hemisphere separately for each subject. Then, the resulting Pearson's correlation coefficients of each subject were Fisher *z*-transformed and analysed using a univariate *t*-test that was significant in both the left (mean $z = 0.84$, s.d. $= 0.56$, $t_{25} = 7.6$, $p < 0.0001$) and right (mean $z = 0.54$, s.d. $= 0.58$, $t_{25} = 4.8$, $p < 0.0001$) hemispheres. This finding confirms that both engravings and objects perceptions recruit the ventral pathway in a similar way in both hemispheres.

The proximity of the ventral visual cortex responses to these two visual categories was likely elicited by the global visual organization of engravings, which triggered processes comparable to those required for object recognition. Among the regions activated by the perception of both engravings and objects, O3-1 and O3-2 corresponded to the so-called LO functional region corresponding to the posterior and lateral part of the lateral occipital complex (LOC) [45]. LO is defined as the brain area showing the greatest activation while viewing a known or novel object compared to its scrambled version [45–47]. As shown in electronic supplementary material, table S3, these two regions exhibited the largest activation in the 'objects *minus* scrambled objects' contrast. The involvement of LO further supports the hypothesis that the processes involved in object recognition are involved in the perception of engravings. This hypothesis is consistent with the fact that LO has been proven to be sensitive to the shape, but not the semantics, of objects [48–52]. Concerning the ventral part of the LOC, includes in the fusiform gyrus, it has been shown that the left fusiform gyrus, which is involved in the visual processing of engravings, is sensitive to semantic information [53,54]. However, the studies mentioned above did not report such a property. Thus, although the similarity between the profiles of objects and engravings does not demonstrate that engravings were used as symbols by their Palaeolithic makers, it is clear that modern brains perceive the engravings as coherent visual entities to which symbolic meaning can be attached. There is of course no guarantee that the brain areas activated by the perception of engravings were, in our ancestors, identical to ours. Regarding the visual cortex, anatomical–functional differences do not probably represent a main bias. Evolution does not seem to have profoundly modified its structure [55,56]. Moreover, investigations on functional homologies between monkeys and humans point to a preservation of major functional subdivisions, at least with regard to low-level visual areas and the ventral pathway [57]. Since these regions appear to have been moderately impacted by the evolution of the brain, it is reasonable to think that the present results also apply to other representatives of the *Homo* lineage. Considering that the

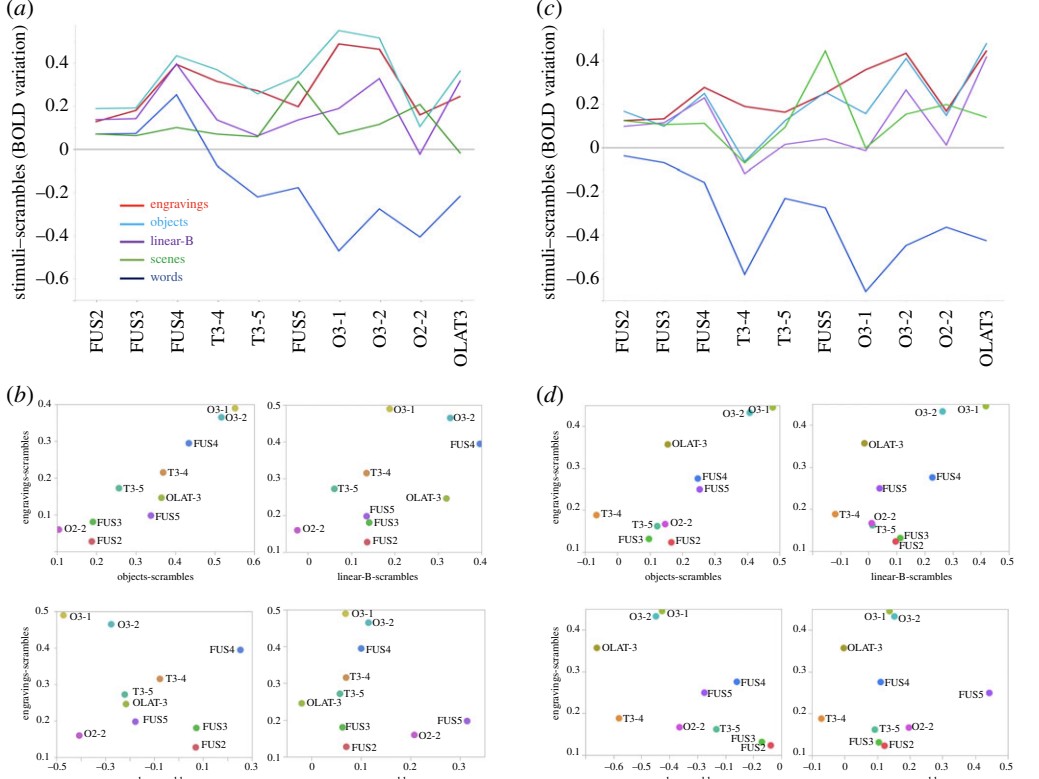

**Figure 3.** Descriptive analysis of the activation profiles of the 10 hROIs of the ventral route activated by engravings perception under the different conditions. Profiles of activity in the left hemisphere (*a*) and right hemisphere (*c*) in regions presented in an antero-posterior sequence along the ventral route. For reasons of readability, the error bars are not shown. They are identical to those shown in figure 2. Plots of the BOLD values obtained under the engravings condition against those obtained under the objects, linear-B, words and scenes conditions across the 10 hROIs in the left (*b*) and right (*d*) hemispheres. The values on the horizontal axis can be positive, null or negative according to the contrast and the hROIs.

engravings were produced deliberately and had no apparent utilitarian function, our results are consistent with the hypothesis that these graphic productions had a representational purpose and were used and perceived as icons or symbols by modern and archaic hominins.

At the regional level, only one area, i.e. the occipito-temporal region of the left hemisphere or FUS4, was activated by both engravings and tasks involving symbolic or iconic contents (objects, linear-B and words). This region, which corresponds to the so-called visual word form area (VWFA) [37,58], was activated by the perception of objects, the linear-B script and words. This is consistent with the view that the role of the left FUS4 is not restricted to word recognition [59–61]. As proposed by Vogel, FUS4 role extends to the processing of complex visual perception with a statistical regularity and a 'groupable' characteristic (i.e. perceived as a whole) [60]. Crucially, our results suggest that these features are found not only in words, images of objects and strings of symbols but also in engravings, supporting their potential representational nature. In addition, rather than focusing on VWFA, we compared the profile of activation of different visual categories in all the regions involved in the perception of the engravings. We found that this profile was similar to that elicited by objects but very different from that elicited by words, VWFA being the only region common to these two categories. Although all participants were literate, they did not process the engravings as potential words, which indicate that literacy was not a confounding factor.

Remarkably, FUS4 was also the only region where the leftward asymmetry survived the FDR correction under the engravings condition (mean = 0.12, $t = 3.17$, $p = 0.04$ FDR corrected, one-sample $t$-test). As clarified by previous studies [38], word presentation causes the largest leftward asymmetry likely due to the hemisphere high-order language areas top-down processing leading to right FUS4 deactivation (table 1). A significant but lower intensity leftward asymmetry was also present during the perception of objects and the linear-B script (figure 4*a*, all $p$'s < 0.05 corrected, one-sample $t$-tests). The leftward asymmetries measured during the engravings, linear-B script and objects presentations did not significantly differ (all $p$'s > 0.10, *post hoc* $t$-tests) but were significantly larger than those

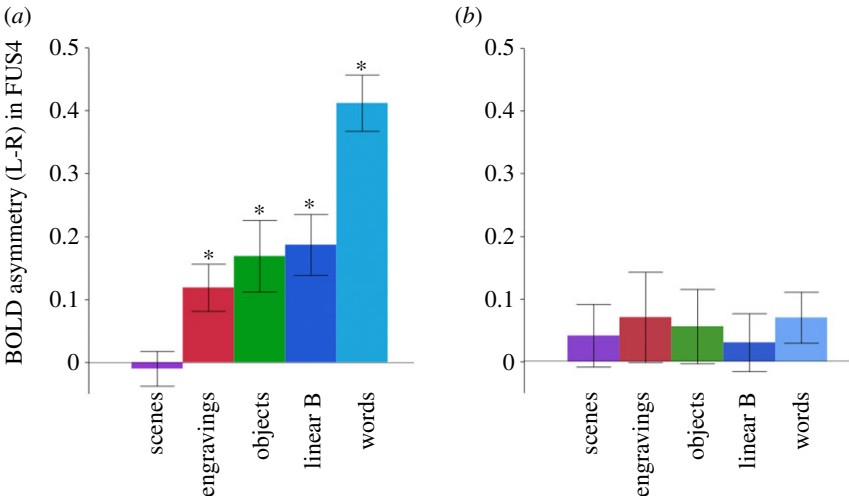

**Figure 4.** Asymmetries of BOLD signal (left *minus* right) in FUS4 under the five conditions. (*a*) Asymmetries measured in the contrast of intact *minus* scrambled stimuli. (*b*) Asymmetries measured during the perception of the scrambled version of each category of stimuli. (*$p < 0.05$, one-sample *t*-test, FDR corrected. Error bars represent the standard error of the mean.)

measured during the scene perception (all $p$'s $< 0.01$) and significantly smaller than those triggered in the words minus scrambled words contrast (all $p$'s $< 0.0001$). To ensure that this result was not determined by variation under the scrambled pictures conditions, we confirmed that the perception of the scrambled pictures of each category did not result, compared to the fixation dot, in any significant asymmetry in this region (see Material and methods, all $p$'s $> 0.10$ uncorrected, figure 4*b*).

The left hemisphere hosts language in most right-handed individuals and is dedicated to cultural artefacts processing in most humans [62]. Under this framework, the leftward asymmetry observed in FUS4 during the perception of engravings suggests that engravings share some representational features with both objects and the linear-B script that are not present in scenes, which did not show any asymmetry ($p = 0.72$, one-sample *t*-test). The trend of gradually decreasing leftward asymmetries in FUS4 from words, objects, linear-B characters, engravings to scenes may reflect the meaning conveyed by these different categories of stimuli, i.e. the more a stimulus is identified as conveying meaning, the more important the leftward asymmetry (figure 4). In sum, since the human brain perceives the engravings as graphic entities having regularities to which semantic information can be connected, our findings support the hypothesis that these engraved patterns could have been used by human cultures of the past to store and transmit coded information.

Although our results do not allow us to reach definitive conclusions on the nature of these representations, they support for the first time with experimental data the hypothesis that they have been used as icons or symbols by both early modern and archaic hominins, as suggested in previous works [6,7,9,22,26,63].

# 2. Material and methods

## 2.1. Participants

The study was approved by the 'Sud-ouest outremer III' local Ethics Committee (no. 2016/63). Twenty-seven healthy right-handed adults (mean age $\pm$ s.d.: 21.7 $\pm$ 2.5 years, 15 women) with no neurological history were included after providing written informed consent to participate in the study. The participants had a mean educational level of 15 years of schooling after first grade. One participant was excluded from the analysis due to a technical problem during the fMRI acquisition.

## 2.2. Data acquisition

### 2.2.1. Stimuli

Our experimental protocol included five categories of stimuli, and each category was compared to its scrambled version (figure 1). All the stimuli used in each category are available as electronic supplementary material, figure S2.

The stimuli of interest, labelled engravings, consisted of 50 tracings of abstract patterns created with stone tools on a variety of media that were discovered at archaeological sites dating from the Lower, Middle and Early Upper Palaeolithic eras (see electronic supplementary material, table S1). We used tracings in which the engraved lines were rendered in white on a grey background and the outline of the object was rendered in light grey. The thickness of the lines on the tracings reflects the width of the engraved lines on the original piece. These stimuli were $700 \times 480$ pixels in size (see electronic supplementary material, figure S2).

The pictures of objects ($256 \times 256$ pixels) consisted of nameable human-made artefacts depicted in different shades of grey (see electronic supplementary material, figure S2). The stimuli were used to determine whether the neural bases of the engravings' perception can be distinguished according to their levels of perceptual organization and the meaning they convey. The lowest level of organization was represented by the scrambled pictures, which were designed to serve as a control for the processing of each condition. These stimuli were constructed by randomly scrambling the other pictures into 32 to 124 squares depending of the initial size of the original picture using Matlab software (MathWorks, Inc., Sherborn, MA, USA). This treatment removes any perceptible organization from the images [42]. These stimuli were included as matched reference stimuli and used to subtract non-specific activations linked to the processing of any type of visual information (stimuli were equal in global luminance and size).

The corpus of linear-B consisted of 50 images of six character strings written in white on a grey background. These images were $653 \times 114$ pixels in size (see electronic supplementary material, figure S2).

The word stimuli were bi-syllabic and composed of six letters in lowercase on a grey background. The word stimuli were nouns referring mostly to a concrete notion (80%). The pictures of words had a size of $256 \times 64$ pixels (see electronic supplementary material, figure S2). The word frequency was $15.5 \pm 3$ (s.d.) according to the lexical data Lexicon 3 [64]. Low-frequency words were chosen since these words activate ventral occipito-temporal regions more strongly than frequent words [65].

The outdoor scenes were $256 \times 256$ pixels pictures of scenes, including the landscapes of mountains, beaches or cities (see electronic supplementary material, figure S2).

Objects and scenes pictures were used in previously published studies [66,67].

## 2.2.2. Functional magnetic resonance imaging acquisition

The imaging was performed using a Siemens Prisma 3 tesla MRI scanner. The structural images were acquired with a high-resolution 3D T1-weighted sequence (TR = 2000 ms, TE = 2.03 ms; flip angle = 8°; 192 slices and 1 mm³ isotropic voxel size). The functional images were acquired with a whole-brain $T_2$*-weighted echo planar image acquisition ($T_2$*-EPI Multiband x6, sequence parameters: TR = 850 ms; TE = 35 ms; flip angle = 56°; 66 axial slices and $2.4 \times 2.4 \times 2.4$ mm isotropic voxel size). The functional images were acquired in three sessions. The experiment presentation was programmed in E-prime software (Psychology Software Tools, Pittsburgh, PA, USA). The stimuli were displayed on a 27″ screen. The participants viewed the stimuli through the rear of the magnet bore via a mirror mounted on the head coil.

## 2.2.3. Experimental protocol

The acquisition was organized in three sessions. The participants performed a 1-back task involving pictures of objects, scenes and words and their scrambled versions in one session.

During the first run, the stimuli consisted of 234 distinct pictures presented in greyscale in a bitmap format belonging to one of the following three categories: scenes, objects and words in their intact and scrambled versions. There were 39 different pictures per category. The block-designed paradigm consisted of 18 blocks lasting 12.75 s (six blocks per category and their scramble version). Each of the 15 stimuli within a block was displayed for 300 ms, including two repetitions, and the participants were asked to detect the stimuli. A fixation point was displayed for 12.75 s every two blocks.

During the second and third runs, the subjects performed the same task and the stimuli were the engravings, linear-B strings and their scrambled version. Each run lasted for 4 min and 12 s with a random order of presentation and comprised 12 experimental blocks of 13.6 s interleaved with seven fixation blocks of 12.75 s. Each experimental block contained 11 stimuli and two repetitions with a 200 ms presentation time for each stimulus and an interstimulus interval of 1037 ms.

The accuracy of correct hits in the one-back task was 86.4% (s.d. 9.9%) of correct detections for the engravings and 81.9% (s.d. 10.7%) for the average of objects, scenes and words.

### 2.2.4. Image preprocessing

Image analysis was performed using SPM12 software. The T1-weighted scans of the participants were normalized to a site-specific template (T1-80TVS) matching the MNI space using the SPM12 'segment' procedure with the default parameters. To correct for subject motion during the fMRI runs, the 192 EPI-BOLD scans were realigned within each run using a rigid-body registration. Then, the EPI-BOLD scans were rigidly registered structurally to the T1-weighted scan. The combination of all registration matrices allowed for the warping of the EPI-BOLD functional scans to the standard space with a trilinear interpolation. Once in the standard space, a 6 mm FWHM Gaussian filter was applied.

### 2.2.5. Regional analysis

Within the selected hROIs, the activation profiles across the regions were calculated for each condition contrast (engravings *minus* scramble, objects *minus* scramble, scenes *minus* scramble, linear-B *minus* scramble and words *minus* scramble) for each of the 26 participants.

The objects, linear-B and scenes were compared to the engravings using repeated-measures 3-way ANOVA in each hemisphere (i.e. 2 (conditions) × 10 (hROIs)). In addition, Pearson's correlation coefficients between the profiles of the engravings and objects conditions were computed for each subject and Fisher *z*-transformed. Then, the 26 Z-scores were analysed using a univariate *t*-test to determine whether the profile of objects was correlated to the profile of engravings. This analysis was conducted in each hemisphere separately.

A list of the regions activated by the objects, words, linear-B and scenes compared to their respective scrambled counterparts is presented in electronic supplementary material, tables S3–S6.

Since a leftward asymmetry in the ventral route is typical of the processing of semantic and verbal material [38,59], we calculated the left minus right BOLD values in the selected hROIs during the engraving presentations to identify the hROIs with significant leftward asymmetries (one-sample *t*-test, FDR corrected). Then, we compared the identified hROI(s) asymmetries across all conditions with one-way ANOVA.

Ethics. The study was approved by the 'Sud-ouest outremer III' local Ethics Committee (no. 2016-A01007-44).

Data accessibility. JMP tables including BOLD signal extracted from hROI as well as asymmetry in hROIs are available at https://datadryad.org/review?doi=doi:10.5061/dryad.m8tf061 [68].

Authors' contributions. E.M., N.T.M., A.M. and F.E. conceived the study. E.M., M.S. and A.M. designed the study. E.M., M.S. and S.C. performed the experiment; E.M., M.S., S.C., N.T.M., M.J. and B.M. carried out the analyses; E.M., N.T.M. and F.E. wrote the manuscript with contributions from A.M., B.M. and G.J. All authors gave final approval for publication.

Competing interests. The authors declare no competing interests.

Funding. This research was supported by a grant from IdEx Bordeaux/CNRS (PEPS 2015). It was also partially supported by the Research Council of Norway through its Centers of Excellence funding scheme (Centre for Early Sapience Behaviour, project number 262618) and by Labex LaScArBx-ANR no. ANR-10-LABX-52. A.M. acknowledges financial support of the Wenner-Gren Foundation for Anthropological Research (2013–2017).

Acknowledgements. The authors thank Ginesis Lab (Labcom Programme 2016, ANR 16LCV2-0006-01) for their help in data management and data processing and Carole Peyrin for kindly providing some of the stimuli included in the experiment.

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
