## [Reviewer comments · Royal Society Open Science]

Review History

RSOS-190086.R0 (Original submission)

Review form: Reviewer 1 (Natalie Uomini)

Is the manuscript scientifically sound in its present form?

Yes

Are the interpretations and conclusions justified by the results?

No

Is the language acceptable?

Yes

Is it clear how to access all supporting data?

No

Do you have any ethical concerns with this paper?

No

Have you any concerns about statistical analyses in this paper?

I do not feel qualified to assess the statistics

Recommendation?

Accept with minor revision (please list in comments)

Comments to the Author(s)

NOTE: I think formatting doesn't appear here, so I've attached my review as a separate PDF.

This paper presents a very original study -- which I believe is the first of its kind -- to study the brain activation of people viewing prehistoric engravings. The authors found similarity between how the brain views engravings and pictures of nameable objects. The topic is very innovative and as such, it does not have any precedents to follow. Therefore, the approach had to be created by the authors. While they combine archaeological background and neuroscience background, they use methods from both disciplines but the combination is new, and thus the way they created the stimuli for the experiments has no comparison in previous literature.

The methodology for neuroimaging follows standard protocols (as for example ones I've used in my previous work), so I believe the experimental design is sound. The choice of stimuli using a scrambled version of each stimulus category is excellent.

In the introduction (page 3), the authors state "If the abstract patterns intentionally engraved by hominins were perceived as structured forms, with a potential meaning, their perception should engage the ventral route cortex". Here I wondered what's the opposite hypothesis? Would it be that the ventral route is NOT engaged? Or would another network be more active?

The first major thing I think is missing in this paper is some discussion about symbolism in hominins. As these results have great implications for the topic, there should be more discussion about it. At least readers should be given a brief introduction to the debates, with some key references. It would be a shame to exclude it because it's what makes this paper so original. It is crucial to define "symbols or icons", a term that's used several times in the paper.

My second main point is that I feel the conclusion is too strong given the lack of discussion about it. "Our findings support the hypothesis that these engraved patterns were used by human cultures of the past to store and transmit coded information" (page 12-13)..... "We conclude that they were probably used as icons or symbols by both modern and archaic hominins."

This paper doesn't contain any discussion of all the big debates around the topic of symbolic capacity in hominins (of which the last author is an expert), therefore the authors cannot make this kind of strong conclusion. The fMRI results themselves (which is all that's discussed in the paper) give only one element of data to this hypothesis. Instead, I suggest to reword as "Our fMRI results lend an element of support to the hypothesis that these engravings could have been used as icons or symbols by both modern and archaic hominins, as was previously suggested by other work (+ cite several references)". Hominin symbolism papers should be mentioned in order to give readers enough background to appreciate the paper, including readers from both archaeology and neuroscience, whether they are familiar with the topic or not.

This issue is related to the archaeological engravings that were chosen, as they are listed in Table S1 - the prehistoric engravings which were used as stimuli contain an extremely wide range of types, eras, styles, geographic origins.... What makes us think we can judge them all the same? Maybe some were used as symbols or some were not. It would be useful to provide images of

what these look like - they could be very different visually, and if they are not visually standardized like the other stimuli, this variation would need to be addressed in the discussion.

Thirdly, I think readers should be told more explicitly about the importance of this study, since it's really a first-time study (wow!!) and it makes a big contribution to 2 disciplines. The authors are too modest and they should make more of that!

On page 9, I might have missed it, but what statistical test was used to compute the first correlation "of their activation profiles in 10 regions, compared to their scrambled version, was computed in each hemisphere separately for each subject."?

Also on page 9, "Then, the resulting Pearson's correlation coefficients of each subject were Fisher z-transformed and analysed using a univariate t-test that was significant in both the left ($t(25) = 7.6, p < 0.0001$) and right ($t(25) = 4.8, p < 0.0001$) hemispheres. This finding confirms that both Engravings and object perception recruit the ventral pathway in a similar way in both hemispheres." Where are these data, please? I perhaps missed them in the text?

METHODS

Several elements of detail are missing from the Methods.

On page 13, "The pictures of objects (256 * 256 pixels) were represented by/consisted of nameable human-made artefacts" Were these pictures taken from a database or created? How was their nameability validated?

On page 15, "Each of the 15 stimuli within a block was displayed for 300 ms, including two repetitions, and the participants were asked to detect the stimuli." How did participants respond? By button press? Was their accuracy of response recorded?

Figure 2 - I very much like this figure because it shows clearly the different areas with color codes. But the error bars are very wide in most cases, so this figure makes me wonder if the Engravings are maybe not so similar to Objects..... the statistics are more convincing.

REF. 8 is missing Author name.

Abstract - hominins is twice misspelled (with M).

Review form: Reviewer 2

Is the manuscript scientifically sound in its present form?

No

Are the interpretations and conclusions justified by the results?

No

Is the language acceptable?

Yes

Is it clear how to access all supporting data?

No

Do you have any ethical concerns with this paper?

No

Have you any concerns about statistical analyses in this paper?

I do not feel qualified to assess the statistics

Recommendation?

Major revision is needed (please make suggestions in comments)

Comments to the Author(s)

Mellet and colleagues conducted a study to test whether Palaeolithic engravings elicits similar representations in the visual pathway to objects and this similarity was indeed found. There is only one engraving shown in the paper (there should be more shown) and based on this one example these engravings look like complex shapes. Therefore, I don't see any reason why they would not elicit responses in visual regions, even the ones higher in the hierarchy, as complex shapes have been shown to elicit responses in these regions. Therefore, I find this question not very novel. I could understand an added value of testing the actual engravings but I can't even evaluate them as they are not shown in the manuscript or in supplementary material. In general, there is not enough information in the manuscript to fully evaluate it and the unreadability of some of the figures makes it hard to evaluate the findings.

I see several issues with this manuscript that I would invite the authors to address.

1. As previously mentioned the stimuli of the engravings are not shown and that seems to be crucial for understanding the value of this paper. Other stimuli classes are also not shown and the authors do not state what categories the objects were sampled from and what words were used. All this information is important for this paper.
2. Why stimulus time presentation is not the same for objects, scenes, and words, and for engravings (300ms vs 200 ms)? If they are being directly compared as few parameters as possible should differ between the conditions.
3. It is not enough to say O3-1 and O3-2 correspond LO. To make this claim the authors should show an overlap of these regions using a LO mask from for example Wang atlas.
4. Different ROIs should be discussed more. It is not enough to say that 10 ROIs elicit a given response pattern. It is important to discuss more what these regions are implicated in and why the result makes sense. The authors try this procedure for LO, however, even there they discuss only a part of the picture. The authors say that "LO is sensitive to the shape, but not the semantics". There is a body of literature that claims otherwise and this should be also discussed.
5. The authors partly acknowledge that they can't claim that these engravings elicited similar patterns of responses as objects in early humans. However, this should be stressed more. The fact that the visual cortex did not expand that much during the evolution does not mean that the visual representations in the cortex in early humans and people nowadays were the same. A [potentially stronger argument could be that the visual representations in humans and macaque monkeys are similar and therefore it is likely that the visual representations of early humans were similar, however, we can't explicitly test that.
6. Resolution of figures is not acceptable in the paper. I can't even read the values on the y-axis of Figure 2 and therefore comment on the results.
7. Labels should be added in panels B and D in Figure 3 as otherwise the dots are not readable.

Minor comments:

1. Table 1 could be presented in a color-coded way to enable rapid detection of significant values, e.g., significant values colored in green.
2. The header should be "funding" not "fundings", as the latter word does not exist in the English language.

Decision letter (RSOS-190086.R0)

24-Apr-2019

Dear Dr Mellet,

The editors assigned to your paper ("Neuroimaging supports the representational nature of the earliest human engravings") have now received comments from reviewers. We would like you to revise your paper in accordance with the referee and Associate Editor suggestions which can be found below (not including confidential reports to the Editor). Please note this decision does not guarantee eventual acceptance.

Please submit a copy of your revised paper before 17-May-2019. Please note that the revision deadline will expire at 00.00am on this date. If we do not hear from you within this time then it will be assumed that the paper has been withdrawn. In exceptional circumstances, extensions may be possible if agreed with the Editorial Office in advance. We do not allow multiple rounds of revision so we urge you to make every effort to fully address all of the comments at this stage. If deemed necessary by the Editors, your manuscript will be sent back to one or more of the original reviewers for assessment. If the original reviewers are not available, we may invite new reviewers.

- Data accessibility

If you wish to submit your supporting data or code to Dryad (<http://datadryad.org/>), or modify your current submission to dryad, please use the following link:
<http://datadryad.org/submit?journalID=RSOS&manu=RSOS-190086>

- Competing interests

- Authors' contributions

- Acknowledgements

- Funding statement

Kind regards,

Andrew Dunn

on behalf of Dr Isabelle Mareschal (Associate Editor) and Essi Viding (Subject Editor)
 openscience@royalsociety.org

Associate Editor's comments (Dr Isabelle Mareschal):

Associate Editor: 1

Comments to the Author:

Expert reviewers have read your paper and raised important questions. Please provide a point by point response to the reviewers, explaining how you have addressed their concerns.

Comments to Author:

Reviewers' Comments to Author:

Reviewer: 1

Comments to the Author(s)

NOTE: I think formatting doesn't appear here, so I've attached my review as a separate PDF.

This paper presents a very original study -- which I believe is the first of its kind -- to study the brain activation of people viewing prehistoric engravings. The authors found similarity between how the brain views engravings and pictures of nameable objects. The topic is very innovative and as such, it does not have any precedents to follow. Therefore, the approach had to be created by the authors. While they combine archaeological background and neuroscience background, they use methods from both disciplines but the combination is new, and thus the way they created the stimuli for the experiments has no comparison in previous literature.

The methodology for neuroimaging follows standard protocols (as for example ones I've used in my previous work), so I believe the experimental design is sound. The choice of stimuli using a scrambled version of each stimulus category is excellent.

In the introduction (page 3), the authors state "If the abstract patterns intentionally engraved by hominins were perceived as structured forms, with a potential meaning, their perception should engage the ventral route cortex". Here I wondered what's the opposite hypothesis? Would it be that the ventral route is NOT engaged? Or would another network be more active?

The first major thing I think is missing in this paper is some discussion about symbolism in hominins. As these results have great implications for the topic, there should be more discussion about it. At least readers should be given a brief introduction to the debates, with some key references. It would be a shame to exclude it because it's what makes this paper so original. It is crucial to define "symbols or icons", a term that's used several times in the paper.

My second main point is that I feel the conclusion is too strong given the lack of discussion about it. "Our findings support the hypothesis that these engraved patterns were used by human cultures of the past to store and transmit coded information" (page 12-13)..... "We conclude that they were probably used as icons or symbols by both modern and archaic hominins." This paper doesn't contain any discussion of all the big debates around the topic of symbolic capacity in hominins (of which the last author is an expert), therefore the authors cannot make this kind of strong conclusion. The fMRI results themselves (which is all that's discussed in the paper) give only one element of data to this hypothesis. Instead, I suggest to reword as "Our fMRI results lend an element of support to the hypothesis that these engravings could have been used as icons or symbols by both modern and archaic hominins, as was previously suggested by

other work (+ cite several references)". Hominin symbolism papers should be mentioned in order to give readers enough background to appreciate the paper, including readers from both archaeology and neuroscience, whether they are familiar with the topic or not.

This issue is related to the archaeological engravings that were chosen, as they are listed in Table S1 - the prehistoric engravings which were used as stimuli contain an extremely wide range of types, eras, styles, geographic origins.... What makes us think we can judge them all the same? Maybe some were used as symbols or some were not. It would be useful to provide images of what these look like - they could be very different visually, and if they are not visually standardized like the other stimuli, this variation would need to be addressed in the discussion.

Thirdly, I think readers should be told more explicitly about the importance of this study, since it's really a first-time study (wow!!) and it makes a big contribution to 2 disciplines. The authors are too modest and they should make more of that!

On page 9, I might have missed it, but what statistical test was used to compute the first correlation "of their activation profiles in 10 regions, compared to their scrambled version, was computed in each hemisphere separately for each subject."?

Also on page 9, "Then, the resulting Pearson's correlation coefficients of each subject were Fisher z-transformed and analysed using a univariate t-test that was significant in both the left ($t(25) = 7.6, p < 0.0001$) and right ($t(25) = 4.8, p < 0.0001$) hemispheres. This finding confirms that both Engravings and object perception recruit the ventral pathway in a similar way in both hemispheres." Where are these data, please? I perhaps missed them in the text?

METHODS

Several elements of detail are missing from the Methods.

On page 13, "The pictures of objects (256 * 256 pixels) were represented by/consisted of nameable human-made artefacts" Were these pictures taken from a database or created? How was their nameability validated?

On page 15, "Each of the 15 stimuli within a block was displayed for 300 ms, including two repetitions, and the participants were asked to detect the stimuli." How did participants respond? By button press? Was their accuracy of response recorded?

Figure 2 - I very much like this figure because it shows clearly the different areas with color codes. But the error bars are very wide in most cases, so this figure makes me wonder if the Engravings are maybe not so similar to Objects..... the statistics are more convincing.

REF. 8 is missing Author name.

Abstract - hominins is twice misspelled (with M).

Reviewer: 2

Comments to the Author(s)

Mellet and colleagues conducted a study to test whether Palaeolithic engravings elicits similar representations in the visual pathway to objects and this similarity was indeed found. There is only one engraving shown in the paper (there should be more shown) and based on this one example these engravings look like complex shapes. Therefore, I don't see any reason why they would not elicit responses in visual regions, even the ones higher in the hierarchy, as complex

shapes have been shown to elicit responses in these regions. Therefore, I find this question not very novel. I could understand an added value of testing the actual engravings but I can't even evaluate them as they are not shown in the manuscript or in supplementary material. In general, there is not enough information in the manuscript to fully evaluate it and the unreadability of some of the figures makes it hard to evaluate the findings.

I see several issues with this manuscript that I would invite the authors to address.

1. As previously mentioned the stimuli of the engravings are not shown and that seems to be crucial for understanding the value of this paper. Other stimuli classes are also not shown and the authors do not state what categories the objects were sampled from and what words were used. All this information is important for this paper.

2. Why stimulus time presentation is not the same for objects, scenes, and words, and for engravings (300ms vs 200 ms)? If they are being directly compared as few parameters as possible should differ between the conditions.

3. It is not enough to say O3-1 and O3-2 correspond LO. To make this claim the authors should show an overlap of these regions using a LO mask from for example Wang atlas.

4. Different ROIs should be discussed more. It is not enough to say that 10 ROIs elicit a given response pattern. It is important to discuss more what these regions are implicated in and why the result makes sense. The authors try this procedure for LO, however, even there they discuss only a part of the picture. The authors say that "LO is sensitive to the shape, but not the semantics". There is a body of literature that claims otherwise and this should be also discussed.

5. The authors partly acknowledge that they can't claim that these engravings elicited similar patterns of responses as objects in early humans. However, this should be stressed more. The fact that the visual cortex did not expand that much during the evolution does not mean that the visual representations in the cortex in early humans and people nowadays were the same. A [potentially stronger argument could be that the visual representations in humans and macaque monkeys are similar and therefore it is likely that the visual representations of early humans were similar, however, we can't explicitly test that.

6. Resolution of figures is not acceptable in the paper. I can't even read the values on the y-axis of Figure 2 and therefore comment on the results.

7. Labels should be added in panels B and D in Figure 3 as otherwise the dots are not readable.

Minor comments:

1. Table 1 could be presented in a color-coded way to enable rapid detection of significant values, e.g., significant values colored in green.

2. The header should be "funding" not "fundings", as the latter word does not exist in the English language.

Author's Response to Decision Letter for (RSOS-190086.R0)

See Appendix A.

RSOS-190086.R1 (Revision)

Review form: Reviewer 1 (Natalie Uomini)

Is the manuscript scientifically sound in its present form?

Yes

Are the interpretations and conclusions justified by the results?

Yes

Is the language acceptable?

Yes

Is it clear how to access all supporting data?

Yes

Do you have any ethical concerns with this paper?

No

Have you any concerns about statistical analyses in this paper?

No

Recommendation?

Accept as is

Comments to the Author(s)

I am satisfied that the authors made good edits in response to the reviewers' comments. I have no further suggestions.

Review form: Reviewer 2

Is the manuscript scientifically sound in its present form?

Yes

Are the interpretations and conclusions justified by the results?

Yes

Is the language acceptable?

Yes

Is it clear how to access all supporting data?

Yes

Do you have any ethical concerns with this paper?

No

Have you any concerns about statistical analyses in this paper?

No

Recommendation?

Accept as is

Comments to the Author(s)

The authors have addressed my comments.

Decision letter (RSOS-190086.R1)

04-Jun-2019

Dear Dr Mellet,

I am pleased to inform you that your manuscript entitled "Neuroimaging supports the representational nature of the earliest human engravings" is now accepted for publication in Royal Society Open Science.

on behalf of Dr Isabelle Mareschal (Associate Editor) and Essi Viding (Subject Editor)
openscience@royalsociety.org

Reviewer comments to Author:

Reviewer: 2

Comments to the Author(s)

The authors have addressed my comments.

Reviewer: 1

Comments to the Author(s)

I am satisfied that the authors made good edits in response to the reviewers' comments. I have no further suggestions.

Appendix A

Response to reviewers

Reviewer 1

Comment: This paper presents a very original study -- which I believe is the first of its kind -- to study the brain activation of people viewing prehistoric engravings. The authors found similarity between how the brain views engravings and pictures of nameable objects. The topic is very innovative and as such, it does not have any precedents to follow. Therefore, the approach had to be created by the authors. While they combine archaeological background and neuroscience background, they use methods from both disciplines but the combination is new, and thus the way they created the stimuli for the experiments has no comparison in previous literature. The methodology for neuroimaging follows standard protocols (as for example ones I've used in my previous work), so I believe the experimental design is sound. The choice of stimuli using a scrambled version of each stimulus category is excellent.

Answer: We thank the reviewer for this positive comment.

Comment: In the introduction (page 3), the authors state "If the abstract patterns intentionally engraved by hominins were perceived as structured forms, with a potential meaning, their perception should engage the ventral route cortex". Here I wondered what's the opposite hypothesis? Would it be that the ventral route is NOT engaged? Or would another network be more active?

Answer: Most of the engravings have a very simple structure, limited to a few lines, sometimes intertwined, sometimes parallel. This relatively simple organization may have not been sufficiently different from the scrambled version to elicit areas along the ventral route. In this case, no activation would have been highlighted by the "engraving - scramble" contrast. Another possibility would have been to find a significant difference between the actual engravings and their scrambled version only in the primary visual cortex, as hypothesized by some authors. The fact that the ventral pathway is activated suggests that the engravings are not perceived as a collection of simple elements but as coherent graphic entities. We conclude that this is a necessary condition for a meaning to be attached to it and for it to be a vector of information.

To introduce the alternative hypothesis, we added the following paragraph in the introduction section (highlighted in red in the revised version of the manuscript): "*The relatively simple organization that characterizes the earliest engravings may be perceived as not being sufficiently different from the scrambled version and may therefore not engage differently brain visual regions. In this case, no activation will be highlighted by the "engraving minus scrambled engravings" contrast. Another possibility is that the difference does not involve the ventral cortex but just the primary visual area. This cortical region has been hypothesized to have played a crucial role in the production of the earliest engravings (Hodgson, 2006; Hodgson, 2014).*"

Comment: The first major thing I think is missing in this paper is some discussion about symbolism in hominins. As these results have great implications for the topic, there should be more discussion about it. At least readers should be given a brief introduction to the debates, with some key references. It would be a shame to exclude it because it's what makes this paper so original. It is crucial to define "symbols or icons", a term that's used several times in the paper.

Answer: We have considerably enlarged the introduction section and added a number of key references to present the current debate on the emergence of symbolic material culture. We have also introduced at the beginning of this section the notion of symbol and provided a definition.

Comment: My second main point is that I feel the conclusion is too strong given the lack of discussion about it. "Our findings support the hypothesis that these engraved patterns were used by human cultures of the past to store and transmit coded information" (page 12-13)..... "We conclude that they were probably used as icons or

symbols by both modern and archaic hominins."

This paper doesn't contain any discussion of all the big debates around the topic of symbolic capacity in hominins (of which the last author is an expert), therefore the authors cannot make this kind of strong conclusion. The fMRI results themselves (which is all that's discussed in the paper) give only one element of data to this hypothesis. Instead, I suggest to reword as "Our fMRI results lend an element of support to the hypothesis that these engravings could have been used as icons or symbols by both modern and archaic hominins, as was previously suggested by other work (+ cite several references)". Hominin symbolism papers should be mentioned in order to give readers enough background to appreciate the paper, including readers from both archaeology and neuroscience, whether they are familiar with the topic or not.

Answer: We have focused the discussion on the fMRI results in order to limit over-interpretations and avoid reaching conclusions based on controversial hypotheses proposed by researchers working in other disciplines rather than on our own results. The state of the debate and the ambiguities implicit in the interpretation of the archaeological record are now presented in the introduction section and we think there is no point in reinjecting them again in the discussion. However, we have softened the conclusion, as suggested by the reviewer. The sentence now reads:

"Although our results do not allow us to reach definitive conclusions on the nature of these representations, they support for the first time with experimental data the hypothesis that they have been used as icons or symbols by both early modern and archaic hominins, as suggested in previous works (d'Errico, 2003; Henshilwood et al., 2009; Rodriguez-Vidal et al., 2014; Villa and Roebroeks, 2014; Majkic et al., 2017; Majkić et al., 2018)"

Comment: This issue is related to the archaeological engravings that were chosen, as they are listed in Table S1 - the prehistoric engravings which were used as stimuli contain an extremely wide range of types, eras, styles, geographic origins.... What makes us think we can judge them all the same? Maybe some were used as symbols or some were not. It would be useful to provide images of what these look like - they could be very different visually, and if they are not visually standardized like the other stimuli, this variation would need to be addressed in the discussion.

Answer: The images of all the engravings are now presented in the supplementary material.

Comment: Thirdly, I think readers should be told more explicitly about the importance of this study, since it's really a first-time study (wow!!) and it makes a big contribution to 2 disciplines. The authors are too modest and they should make more of that!

Answer: We have added a sentence in the introduction section in which we underline the novelty of the study. *"We report here the first attempt to shed light on the function of Paleolithic engravings by mapping the brain regions involved in their perception"*.

Comment: On page 9, I might have missed it, but what statistical test was used to compute the first correlation "of their activation profiles in 10 regions, compared to their scrambled version, was computed in each hemisphere separately for each subject."?

Answer: In that section, we raise the question as to whether there a significant relationship between the profile of activation during engravings and objects perception when individual variability is considered. We first computed a Pearson's correlation between these two profiles of activation in each subject. This produced 26 x 2 hemispheres Pearson's r coefficients (one per participant and per hemisphere). No statistical test were performed at this stage since the distribution of r coefficient does not allow to use a parametric test (see below).

Comment: Also on page 9, "Then, the resulting Pearson's correlation coefficients of each subject were Fisher z -transformed and analysed using a univariate t -test that was significant in both the left ($t(25) = 7.6, p < 0.0001$) and right ($t(25) = 4.8, p < 0.0001$) hemispheres. This finding confirms that both Engravings and object perception recruit the ventral pathway in a similar way in both hemispheres." Where are these data, please? I perhaps missed them in the text?

Answer: The Fisher-z transformation of the r coefficients allows to perform a univariate t-test on the z-scores to assess whether they significantly differed from zero in average. We did not include the 52 z-scores in the results but only the result of the t-test for each hemisphere. In the revised version of the manuscript we added the mean and standard deviation for the two hemispheres (left hemisphere: mean z= 0.84, SD=0.56, right hemisphere mean z= 0.54, SD=0.58).

METHODS

Several elements of detail are missing from the Methods.

Comment: On page 13, "The pictures of objects (256 * 256 pixels) were represented by/consisted of nameable human-made artefacts" Were these pictures taken from a database or created? How was their nameability validated?

Response: The objects and scenes stimuli were kindly provided by another team and come from previous published studies (Kauffmann et al., 2015) (Roux-Sibilon et al., 2018) (Kauffmann et al, 2015, Roux-Sibilon et al, 2018). The pictures come from the Microsoft database and are now included in the supplementary material. In the revised version of the manuscript we add the references of these studies. Although nameability was not formally evaluated the chosen objects fall in easily namable categories.

Comments: On page 15, "Each of the 15 stimuli within a block was displayed for 300 ms, including two repetitions, and the participants were asked to detect the stimuli." How did participants respond? By button press? Was their accuracy of response recorded?

Response: Yes, the participants responded by button press each time a stimulus was repeated. The accuracy was recorded. It was 86.4 % (SD 9.9%) of correct detections for the engravings and 81.9% (SD 10.7%) for the average of objects, scenes and words. We have added this information in the revised manuscript.

Comment: Figure 2 - I very much like this figure because it shows clearly the different areas with color codes. But the error bars are very wide in most cases, so this figure makes me wonder if the Engravings are maybe not so similar to Objects..... the statistics are more convincing.

Response: We agree with this reviewer that the similarity is better supported by the statistics. The purpose of the figure is just that of summarizing the information in easy-to-read format. There are two arguments in favour of a similarity between engravings and objects: the absence of interaction conditions x hROIS for these two conditions (whereas the interaction exists between engravings and the other conditions) and the existence of a significant correlation between the activation profile of engravings and objects.

Comment: REF. 8 is missing Author name.

Response: It has been corrected.

Comment: Abstract - hominins is twice misspelled (with M).

Response: It has been corrected.

Reviewer2:

Comment: Mellet and colleagues conducted a study to test whether Paleolithic engravings elicits similar representations in the visual pathway to objects and this similarity was indeed found. There is only one engraving shown in the paper (there should be more shown) and based on this one example these engravings look like complex shapes. Therefore, I don't see any reason why they would not elicit responses in visual regions, even the ones higher in the hierarchy, as complex shapes have been shown to elicit responses in these regions. Therefore, I find this question not very novel. I could understand an added value of testing the actual engravings but I can't

even evaluate them as they are not shown in the manuscript or in supplementary material. In general, there is not enough information in the manuscript to fully evaluate it and the unreadability of some of the figures makes it hard to evaluate the findings.

Response: To be precise, the goal of our research was not that of contrasting the perception of Paleolithic engravings and objects. It involved contrasting the earliest engravings with four visual categories, including objects, and their scrambled version, in order to evaluate what areas were more elicited by the earliest engravings, which present different degrees of complexity. Although we concur with this reviewer that involvement of areas higher in the hierarchy was a reasonable expectation, at least for the more complex engravings, the only available and highly widespread theory before our research to explain the emergence of this behaviour in the evolution of our genus (Hodgson, 2006, 2014) was predicting that the emergence and perception of these engravings exclusively involved the primary visual cortex. In the revised version of the manuscript, we explain previous hypotheses in more detail and present the images of all objects and the tracings of all the engravings included in the experiments. The resolution of the figures has also been improved. As a result of these changes, the novelty of the study is now more apparent and the dataset more explicit. Testing the photos of the actual objects would have introduced a bias in the research since engravings occur on media of different colour, texture and state of preservation.

I see several issues with this manuscript that I would invite the authors to address.

Comment: As previously mentioned the stimuli of the engravings are not shown and that seems to be crucial for understanding the value of this paper. Other stimuli classes are also not shown and the authors do not state what categories the objects were sampled from and what words were used. All this information is important for this paper.

Response: We thank the reviewer for this suggestion. All the stimuli used are now provided as supplementary material.

Comment: Why stimulus time presentation is not the same for objects, scenes, and words, and for engravings (300ms vs 200 ms)? If they are being directly compared as few parameters as possible should differ between the conditions.

Response: The conditions including the engravings and strings of linear B characters were the subject of a behavioural pre-manipulation that allowed us to optimize the presentation times for these stimuli. The stimuli for the other conditions were provided by another team and came from previous published studies (Kauffmann et al, 2015, Roux-Sibilon et al, 2018) for which presentation times were 100 ms longer. Since the two presentation times resulted from a well-argued choice, we preferred not to modify them. We are aware that it is preferable to limit the differences but it is important to note that the visual categories were not compared directly but that we compared the difference between intact stimuli and their scrambled versions (whose presentation times were identical). We only compare the visual categories on the basis of this difference. The potential biases related to the difference in presentation time are therefore eliminated.

Comment: It is not enough to say O3-1 and O3-2 correspond LO. To make this claim the authors should show an overlap of these regions using a LO mask from for example Wang atlas.

Response: We plotted the maximum activation of several studies which located LO, and superimposed them on the O3-1 and O3-2 regions. As one can see here below these peaks project quite well on these two regions (yellow: O3-1, blue: O3-2). given that the SD for the peaks coordinates vary from 7 to 10 mm.

In addition, it is in these two regions that the activation is maximum in the contrast “object *minus* scrambled object” from the present study. This contrast is typically used to locate the LOC. We added the following sentence in the revised version of the manuscript: “LO is defined as the brain area showing the greatest activation while viewing a known or novel object compared to its scrambled version (Malach et al., 1995; Grill-Spector et al., 2001). As shown in table S3 (supplementary material) these two regions exhibited the largest activation in the “objects *minus* scrambled objects” contrast in the present study.”

Comment: Different ROIs should be discussed more. It is not enough to say that 10 ROIs elicit a given response pattern. It is important to discuss more what these regions are implicated in and why the result makes sense. The authors try this procedure for LO, however, even there they discuss only a part of the picture. The authors say that “LO is sensitive to the shape, but not the semantics”. There is a body of literature that claims otherwise and this should be also discussed.

Response:

Besides the ventral regions for which a specificity has been shown (such as LOC, VWFA, FFA etc...), many regions have not yet been specifically involved in the processing of a particular visual category. It is possible that these regions have a more general purpose (Grill-Spector, 2003). It has also been proposed that the representation of a percept is reflected by a distinct pattern of response across all ventral cortex, and this distributed activation produces the visual perception (Haxby et al., 2001). In this context, it is difficult to discuss the involvement of each region and we preferred to discuss the overall activation pattern along the ventral pathway. Nevertheless, when some regions corresponded to well documented functional areas (LOC, VWFA...) we discussed them in accordance with the existing literature.

The reviewer raised an important issue regarding the role of LO in semantic. To our knowledge, there is no study that has reported such sensitivity in the lateral occipital (LO) part of the LOC. On the contrary, several studies have shown that activity in this region (mainly based on adaptation paradigm) is not affected by the change in visual categories (Grill-Spector et al., 1999; Vuilleumier et al., 2002; Chouinard et al., 2008; Kim et al., 2009; Margalit et al., 2017). The situation is more nuanced with regard to the ventral part of the LOC (called Pfs). At least two studies reported an effect of visual categories in the left fusiform gyrus (Koutstaal et al., 2001; Simons et al., 2003). However, none of the studies cited above reported this activation and they concluded that the entire LOC is not sensitive to semantic information. We added the following sentence in the revised manuscript: “Concerning the ventral part of the LOC, it has been shown that the left fusiform gyrus, which is involved in the visual processing of engravings, is sensitive to semantic information (Koutstaal et al., 2001; Simons et al., 2003). However, the studies mentioned above did not report such a property”.

Comment: The authors partly acknowledge that they can’t claim that these engravings elicited similar patterns of responses as objects in early humans. However, this should be stressed more. The fact that the visual cortex did not

expand that much during the evolution does not mean that the visual representations in the cortex in early humans and people nowadays were the same. A [potentially stronger argument could be that the visual representations in humans and macaque monkeys are similar and therefore it is likely that the visual representations of early humans were similar, however, we can't explicitly test that.

Response: The reviewer is right in pointing out the main difficulty of our approach. Inferring the cognitive abilities of fossil human populations from the functional study of the modern human brain is, of course, not a simple undertaking. As suggested by the reviewer regarding the visual cortex, anatomical-functional differences probably do not represent a main bias. Evolution does not seem to have profoundly modified its structure (Ponce de León et al., 2016; Holloway et al., 2018). The study of functional homologies between monkeys and humans points to a preservation of major functional subdivisions, at least with regard to low-level visual areas and the ventral pathway (Orban et al., 2004). As pointed out by the reviewer, this does not guarantee with certainty that the representations in the visual cortex of our ancestors were identical to ours. The relatively small impact of evolution on this cortex suggests that inferences about the past can reasonably be made from results obtained when working with the modern brain. Another aspect that needs to be considered is that if these engravings were familiar to the past hominins, who purposely produced them, they were not to our participants. This may have an effect on the neural networks mobilized. We are currently conducting the same study with archaeologists who are experts in this type of material in order to compensate for this familiarity bias.

The following paragraph has been added at page 8 (in red in the revised version): *“There is of course no guarantee that the brain areas activated by the engravings were, in our ancestors, identical to ours. Regarding the visual cortex, anatomical-functional differences do not probably represent a main bias. Evolution does not seem to have profoundly modified its structure (Ponce de León et al., 2016; Holloway et al., 2018). Moreover, investigations on functional homologies between monkeys and humans points to a preservation of major functional subdivisions, at least with regard to low-level visual areas and the ventral pathway (Orban et al., 2004). Since these regions appear to have been moderately impacted by the evolution of the brain it is reasonable to think that the present results also apply to other representatives of the Homo lineage”.*

Comment: Resolution of figures is not acceptable in the paper. I can't even read the values on the y-axis of Figure 2 and therefore comment on the results.

Response: We have improved the resolution of the figures and increased the size of the characters for a better readability.

Comment: Labels should be added in panels B and D in Figure 3 as otherwise the dots are not readable.

Response: As requested by the reviewer, we added the labels in all the panels of the Figure 3.

Minor comments:

Comment: Table 1 could be presented in a color-coded way to enable rapid detection of significant values, e.g., significant values colored in green.

Response: Table 1 now includes a color code to display significant activation, deactivation and left-right asymmetries.

Comment: The header should be “funding” not “fundings”, as the latter word does not exist in the English language.

Response: It has been corrected

References:

- Chouinard, PA, Morrissey, BF, Köhler, S, Goodale, MA (2008) Repetition suppression in occipital-temporal visual areas is modulated by physical rather than semantic features of objects. *Neuroimage*, 41:130–144.
- d’Errico, F (2003) The invisible frontier. A multiple species model for the origin of behavioral modernity. *Evolution Anthropology*, 12:188–202.

- Grill-Spector, K (2003) The neural basis of object perception. *Curr.Opin.Neurobiol.*, 13:159–166.
- Grill-Spector, K, Kushnir, T, Edelman, S, Avidan, G, Itzhak, Y, Malach, R (1999) Differential processing of objects under various viewing conditions in the human lateral occipital complex. *Neuron*, 24:187–203.
- Grill-Spector, K, Kourtzi, Z, Kanwisher, N (2001) The lateral occipital complex and its role in object recognition. *Vision research*, 41:1409–1422.
- Haxby, JV, Gobbini, MI, Furey, ML, Ishai, A, Schouten, JL, Pietrini, P (2001) Distributed and overlapping representations of faces and objects in ventral temporal cortex. *Science*, 293:2425–2430.
- Henshilwood, CS, d’Errico, F, Watts, I (2009) Engraved ochres from the middle stone age levels at Blombos Cave, South Africa. *Journal of human evolution*, 57:27–47.
- Hodgson, D (2006) Understanding the origins of Paleoaart: the neurovisual resonance theory and brain functioning. *Paleoanthropology*, 2006:54–67.
- Hodgson, D (2014) Decoding the Blombos engravings, shell beads and Diepkloof ostrich eggshell patterns. *Cambridge Archaeological Journal*, 24:57–69.
- Holloway, RL, Hurst, SD, Garvin, HM, Schoenemann, PT, Vanti, WB, Berger, LR, Hawks, J (2018) Endocast morphology of *Homo naledi* from the Dinaledi Chamber, South Africa. *Proc Natl Acad Sci U S A*, 115:5738–5743.
- Kauffmann, L, Ramanoël, S, Guyader, N, Chauvin, A, Peyrin, C (2015) Spatial frequency processing in scene-selective cortical regions. *Neuroimage*, 112:86–95.
- Kim, JG, Biederman, I, Lescroart, MD, Hayworth, KJ (2009) Adaptation to objects in the lateral occipital complex (LOC): shape or semantics. *Vision Res*, 49:2297–2305.
- Koutstaal, W, Wagner, AD, Rotte, M, Maril, A, Buckner, RL, Schacter, DL (2001) Perceptual specificity in visual object priming: functional magnetic resonance imaging evidence for a laterality difference in fusiform cortex. *Neuropsychologia*, 39:184–199.
- Majkić, A, d’Errico, F, Milošević, S, Mihailović, D, Dimitrijević, V (2018) Sequential incisions on a cave bear bone from the Middle Palaeolithic of Pešturina cave, Serbia. *Journal of Archeological Method and Theory*, 25:69–116.
- Majkic, A, Evans, S, Stepanchuk, V, Tsvelykh, A, d’Errico, F (2017) A decorated raven bone from the Zaskalnaya VI (Kolosovskaya) Neanderthal site, Crimea. *PloS one*, 12:e0173435.
- Malach, R, Reppas, JB, Benson, RR, Kwong, KK, Jiang, H, Kennedy, WA, Ledden, PJ, Brady, TJ, Rosen, BR, Tootell, RB (1995) Object-related activity revealed by functional magnetic resonance imaging in human occipital cortex. *Proc Natl Acad Sci U S A*, 92:8135–8139.
- Margalit, E, Biederman, I, Tjan, BS, Shah, MP (2017) What Is Actually Affected by the Scrambling of Objects When Localizing the Lateral Occipital Complex? *J Cogn Neurosci*, 29:1595–1604.
- Orban, GA, Van Essen, D, Vanduffel, W (2004) Comparative mapping of higher visual areas in monkeys and humans. *Trends Cogn Sci*, 8:315–324.
- Ponce de León, MS, Bienvenu, T, Akazawa, T, Zollikofer, CP (2016) Brain development is similar in Neanderthals and modern humans. *Curr Biol*, 26:R665–6.
- Rodríguez-Vidal, J, d’Errico, F, Giles Pacheco, F, Blasco, R, Rosell, J, Jennings, RP, Queffelec, A, Finlayson, G, Fa, DA, Gutiérrez López, JM, Carrión, JS, Negro, JJ, Finlayson, S, Cáceres, LM, Bernal, MA, Fernández Jiménez, S, Finlayson, C (2014) A rock engraving made by Neanderthals in Gibraltar. *Proc Natl Acad Sci U S A*, 111:13301–13306.
- Roux-Sibilon, A, Kalénine, S, Pichat, C, Peyrin, C (2018) Dorsal and ventral stream contribution to the paired-object affordance effect. *Neuropsychologia*, 112:125–134.
- Simons, JS, Koutstaal, W, Prince, S, Wagner, AD, Schacter, DL (2003) Neural mechanisms of visual object priming: evidence for perceptual and semantic distinctions in fusiform cortex. *NeuroImage*, 19:613–626.
- Villa, P, Roebroeks, W (2014) Neandertal demise: an archaeological analysis of the modern human superiority complex. *PLoS one*, 9:e96424.
- Vuilleumier, P, Henson, RN, Driver, J, Dolan, RJ (2002) Multiple levels of visual object constancy revealed by event-related fMRI of repetition priming. *Nat Neurosci*, 5:491–499.